# Characterization of Tumor and Immune Tumor Microenvironment of Primary Tumors and Metastatic Sites in Advanced Renal Cell Carcinoma Patients Based on Response to Nivolumab Immunotherapy: Preliminary Results from the Meet-URO 18 Study

**DOI:** 10.3390/cancers15082394

**Published:** 2023-04-21

**Authors:** Sara Elena Rebuzzi, Matteo Brunelli, Francesca Galuppini, Valerio Gaetano Vellone, Alessio Signori, Fabio Catalano, Alessandra Damassi, Gabriele Gaggero, Pasquale Rescigno, Marco Maruzzo, Sara Merler, Francesca Vignani, Alessia Cavo, Umberto Basso, Michele Milella, Olimpia Panepinto, Manlio Mencoboni, Marta Sbaraglia, Angelo Paolo Dei Tos, Veronica Murianni, Malvina Cremante, Miguel Angel Llaja Obispo, Michele Maffezzoli, Giuseppe Luigi Banna, Sebastiano Buti, Giuseppe Fornarini

**Affiliations:** 1Medical Oncology Unit, Ospedale San Paolo, 17100 Savona, Italy; 2Department of Internal Medicine and Medical Specialties (Di.M.I.), University of Genoa, 16132 Genoa, Italy; 3Pathology Unit, Department of Diagnostics and Public Health, University and Hospital Trust of Verona, 37124 Verona, Italy; 4Surgical Pathology Unit, Department of Medicine (DIMED), University of Padua, 35128 Padua, Italy; 5Pathology Unit, IRCCS Istituto Giannina Gaslini, 16147 Genoa, Italy; 6Department of Health Sciences (DISSAL), Section of Biostatistics, University of Genoa, 16132 Genoa, Italy; 7Medical Oncology Unit 1, IRCCS Ospedale Policlinico San Martino, 16132 Genoa, Italy; 8Candiolo Cancer Institute, FPO-IRCCS, 10060 Candiolo, Italy; 9Translational and Clinical Research Institute, Centre for Cancer, Newcastle University, Newcastle upon Tyne NE1 7RU, UK; 10Oncology Unit 1, Istituto Oncologico Veneto IOV-IRCCS, 35128 Padua, Italy; 11Section of Oncology, Department of Medicine, University of Verona and Verona University Hospital Trust, 37134 Verona, Italy; 12Division of Medical Oncology, Ordine Mauriziano Hospital, 10128 Turin, Italy; 13Oncology Unit, Villa Scassi Hospital, 16149 Genoa, Italy; 14Department of Medicine and Surgery, University of Parma, 43126 Parma, Italy; 15Department of Oncology, Portsmouth Hospitals University NHS Trust, Portsmouth PO6 3LY, UK; 16Medical Oncology Unit, University Hospital of Parma, 43126 Parma, Italy

**Keywords:** renal cell carcinoma, immunotherapy, immune checkpoint inhibitor, nivolumab, tumor microenvironment, immunohistochemistry, immune infiltration, CD56, lymphocyte

## Abstract

**Simple Summary:**

The identification of clinical and tumor factors to identify patients who most benefit from oncological treatments is a crucial clinical unmet need. We studied differences in the tumor microenvironment assessed using an easy method called immunohistochemistry between cancers in patients who did or did not respond to an immunotherapy called nivolumab. The tumors in those patients who most benefitted from nivolumab were characterized by the following: a poor presence of specific immune cells (CD4+), which are likely to have an immunosuppressive role; a high expression of the CD56 marker on tumor cells, which plays a role in cell cytotoxicity; and a high expression of the phosphorylated form of the mTOR protein in tumor cells, which regulates the function of intra-tumor inflammatory cells and cancer cells. Further immunohistochemical and genomic analyses are planned to deeply examine the prognostic role of the tumor microenvironment.

**Abstract:**

Background: Prognostic and predictive factors for patients with metastatic renal cell carcinoma (mRCC) treated with immunotherapy are highly warranted, and the immune tumor microenvironment (I-TME) is under investigation. Methods: The Meet-URO 18 was a multicentric retrospective study assessing the I-TME in mRCC patients treated with ≥2nd-line nivolumab, dichotomized into *responders* and *non-responders* according to progression-free survival (≥12 months and ≤3 months, respectively). The primary objective was to identify differential immunohistochemical (IHC) patterns between the two groups. Lymphocyte infiltration and the expressions of different proteins on tumor cells (CD56, CD15, CD68, and ph-mTOR) were analyzed. The expression of PD-L1 was also assessed. Results: A total of 116 tumor tissue samples from 84 patients (59% were primary tumors and 41% were metastases) were evaluated. Samples from *responders* (N = 55) were significantly associated with lower expression of CD4+ T lymphocytes and higher levels of ph-mTOR and CD56+ compared with samples from *non-responders* (N = 61). *Responders* also showed a higher CD3+ expression (*p* = 0.059) and CD8+/CD4+ ratio (*p* = 0.084). *Non-responders* were significantly associated with a higher percentage of clear cell histology and grading. Conclusions: Differential IHC patterns between the tumors in patients who were *responders* and *non-responders* to nivolumab were identified. Further investigation with genomic analyses is planned.

## 1. Introduction

In recent years, the treatment landscape of metastatic renal cell carcinoma (mRCC) has been revolutionized with the approval of immune checkpoint inhibitors (ICIs) as monotherapy for pretreated patients and in combination therapies for untreated patients [1]. In 2015, nivolumab was the first FDA-approved ICI after its survival advantage over everolimus, as reported in the Checkmate 025 study [2]. Despite the survival advantage observed with ICI-based therapies, only a small percentage of patients respond to immunotherapy or maintain a durable clinical benefit over time [3]. The identification of prognostic and/or predictive biomarkers is crucial for therapeutic selection and sequencing, maximizing efficacy, sparing patients from unnecessary toxicities, and minimizing costs [3].

Prognostic and/or predictive factors, which are under investigation in mRCC patients treated with immunotherapies, include peripheral blood inflammatory indices, clinical factors, and tumor microenvironment (TME) biomarkers [3,4]. More recently, new frontiers in the investigation of predictive biomarkers for immunotherapy, such as mitochondrial metabolism, have been explored and are of high interest [5].

On the other hand, peripheral blood inflammatory indices, clinical factors, and their combinations in clinical models are of great interest because of their low cost and are readily available and thus easily integrated into therapeutic decision making [6].

The TME is heterogeneous and consists of many types of immune cells (lymphocytes, macrophages, and neutrophils), stromal cells, and the activation of different molecular pathways [4]. Numerous studies have shown the association of the immune compartment of the TME (I-TME), which includes tumor-infiltrating lymphocytes (TILs), tumor-associated macrophages (TAMs), and neutrophils (TANs), with prognoses and treatment responses, including ICIs. Nevertheless, the I-TME is far from being routinely used [4,7,8]. Recently, efforts have been made to implement the understanding of the I-TME and responses to immunotherapy, also considering genomic and transcriptomic signatures [9,10].

The multicenter retrospective Meet-URO 15 study was a large study assessing the prognostic role of peripheral blood inflammatory indices and clinical factors, leading to the development of a new prognostic model (the Meet-URO score) [11]. These results led to the idea of focusing on the differences in the I-TME between patients who most (i.e., *responders*) and least (i.e., *non-responders*) benefited from immunotherapy with nivolumab. We, therefore, derived a follow-up study (the Meet-URO 18 study) assessing the I-TMEs of *responders* and *non-responders* patients performing immunohistochemical (IHC) and molecular analyses. Herein, we report the IHC analyses of this study, while the molecular analyses are still ongoing and will be presented separately.

## 2. Materials and Methods

The Meet-URO 18 study was a multicentric retrospective translational study designed to assess the prognostic role of the I-TME of primary tumors and metastases in patients with mRCC treated with ≥second-line nivolumab and divided into two cohorts according to clinical benefit from immunotherapy.

This study was approved by the Regional Ethical Committee (Regional Ethical Committee of Liguria; registration number: 209/2020-DB ID 10531). The first analyses in this study included 7 Italian centers. They were performed according to the Declaration of Helsinki, Good Clinical Practice, and local ethical guidelines. Written informed consent was signed by all living patients enrolled in this study.

### 2.1. Study Population

The main inclusion criteria included the following: histological diagnosis of renal cell carcinoma; advanced disease; at least one completed infusion of nivolumab given as standard clinical practice and as a second or further treatment line; progression-free survival ≤3 months (*non-responders*) or ≥12 months (*responders*); availability of sufficient histological material of primary tumor and/or metastasis to perform immunohistochemical and molecular analyses; and availability of pre-treatment complete blood count values and the “International Metastatic RCC Database Consortium” (IMDC) score.

### 2.2. Study Procedures

The IHC analyses included the grading, histological revision, and digital multitarget analysis of lymphocyte infiltration and tumor cells (TCs).

The histological revision was performed according to the last WHO 2016 classification of RCC morphology reassessing and immunohistochemical and molecular characteristics. The qualitative and quantitative analysis of lymphocyte infiltration included the morphological and immunophenotypic evaluation of tumor-infiltrating lymphocytes (TILs) within the tumor and at the tumor margin, including CD8+, CD4+, and FOXP3+ T cells, the CD8+/CD4+ ratio, and peri/intra-tumoral T cells.

Assessment of the expressions of CD56, CD15, CD68, and phosphorylated mTOR (ph-mTOR) in TCs was performed. The expression of PD-L1 (SP263) staining in both TCs and tumor-infiltrating immune cells (ICs) was also assessed.

Percentages of immunoreactive cells were counted and aligned to a ×200 (0.933 mm^2^) microscopic field.

### 2.3. Study Objectives

The primary objective of this study was the identification of differential IHC and molecular patterns in the I-TMEs between responder and non-responder patients treated with nivolumab. Secondary objectives included investigating the correlation of the IHC and molecular patterns in the I-TMEs between primary tumors and metastases to identify potential inter-tumor heterogeneity. The present analysis refers only to the primary objective.

### 2.4. Statistical Analysis

The I-TME parameters were evaluated according to the median plus interquartile range (IQR), and the cut-offs were identified via the receiver operating curves (ROCs) based on the PFS of the whole sample.

The chi-square test or the Fisher test was used for percentage comparisons, while Student’s *t*-test or the Mann–Whitney test, when appropriate, was used for the comparisons of the means and the medians, respectively. Statistical significance was defined as *p* < 0.05. All statistical analyses were performed using the software Stata v.16 (StataCorp., College Station, TX, USA, 2019).

## 3. Results

### 3.1. Patients’ Characteristics

A total of 84 patients with mRCC were included in the analysis. The patients’ characteristics are summarized in Table 1.

Most patients were male (73.8%), and the median age was 69 years. A total of 77.4% of patients had clear cell histology, and most of the patients had previously undergone nephrectomy (86.1%). From December 2015 to July 2022, 71.4% of patients started nivolumab as a 2nd-line treatment and 28.6% as a ≥3rd-line one. According to the IMDC risk scores at the time of nivolumab treatment, 21.3% of patients had favorable risk, while 61.3% and 17.5% had intermediate or poor risk. Moreover, lymph node metastases were present in 51.3% (N = 41) of patients, visceral metastasis in 88.8% (N = 71), and bone metastasis in 50% (N = 40).

Notably, non-responder patients were significantly younger (*p* = 0.004) and had a worse prognosis according to the IMDC risk score (*p* = 0.030) compared with responder patients.

### 3.2. IHC Analysis of Primary Tumors and Metastases (Responders vs. Non-Responders)

Overall, 116 tumor tissue samples (59% of which were primary tumors and 41% metastases) were assessed. The overall results are shown in Table 2.

Responder patients were significantly associated with lower expression of CD4+ T lymphocytes (<70: 74.6% vs. 52.5%; *p* = 0.015), higher levels of ph-mTOR (≥15: 70.9% vs. 50.8%; *p* = 0.029), and CD56+ in TCs (≥40: 27.3% vs. 11.5%; *p* = 0.035) compared with non-responder patients (Figure 1).

Responders also showed a numerical difference in CD3 expression (≥40: 72.7% vs. 55.7%; *p* = 0.059) and CD8+/CD4+ ratios (median 1.74 vs. 1.20; *p* = 0.084) in ICs compared with non-responder patients.

Non-responder patients presented a higher percentage of clear cell histology (ccRCC) (85.3% vs. 61.8%; *p* = 0.005) and higher tumors gradings (G3–4: 80.4% vs. 51.4%; *p* < 0.05).

No differences between responders and non-responders were observed according to CD8+ and peritumoral T lymphocytes, the expression of CD15 and CD68 in TCs, and PD-L1 expression.

The IHC assessment of FOXP3+ T cells and CD56+ NK cells showed high heterogeneity in single tumor expression between pathologists; thus, no interpretation and clinical pathological correlation was performed due to a lack of robustness in categorization reporting.

Moreover, according to the ROCs, we were not able to identify an optimal cutoff for the expression of CD8+ T lymphocytes and CD68.

## 4. Discussion

The personalization of cancer therapy is a crucial clinical issue that drives the preclinical and clinical assessment of prognostic and predictive factors in mRCC patients treated with immunotherapy [12]. Clinical factors have the advantage of being easily assessable and applicable, but great interest and effort have been spent on translational research [4]. In this context, the analysis of the composition of the I-TME is essential to understand tumor immunology, which is ultimately involved in the response to these treatment breakthroughs.

Tumor-infiltrating immune cells (TI-ICs) have a central role in pro- and anti-tumorigenic processes and correlate with clinical outcomes, in many cancer types, and responses to different types of treatments, including immunotherapies [13].

This is true for RCC especially, which is among the most immune and vascularly infiltrated cancer types [4]. Therefore, there is great interest in understanding the I-TME in advanced cancer patients treated with immunotherapy, and we reported a comprehensive evaluation of pre-treatment tumor-intrinsic immune cell infiltration in mRCC patients assessed according to the clinical benefit of immunotherapy.

To identify consistent I-TME differences that may be correlated with responses to immunotherapy, the extremes of the real-world context were explored: the patients with mRCC who highly benefited from nivolumab immunotherapy with a PFS of >12 months (*responders*) and those who did not have a PFS of <3 months (*non-responders*). In our analysis, different aspects of the I-TME and tumors were assessed: TI-ICs, the mTOR pathway, tumor histotype, and grading.

The most studied TI-ICs are TILs, which have been associated with the prognosis and response to different types of therapies [14]. More recently, an increased lymphocyte infiltrate was shown to be a positive prognostic factor and a potential predictor of responses to ICIs, including in mRCC patients [15].

Generally, CD8+ cytotoxic T cells and T helper 1 (Th1) CD4+ T cells promote anti-tumor immunity, while regulatory CD4+ T cells (Treg) and T helper 2 (Th2) CD4+ T cells are associated with immune evasion [4]. RCC is generally characterized by rich intra-tumoral T cell infiltration compared with other cancer types; however, contrasting with other tumors, increased CD8+ infiltration is often found to confer a worse prognosis due to the possibly high infiltration of exhausted CD8+ and immunosuppressive TILs [4,16].

In our analysis, *responder* patients were significantly associated with lower CD4+ T lymphocyte expression and showed higher CD3+ T lymphocyte expression and CD8+/CD4+ ratios. These results could possibly suggest the characterization of these CD4+ T cells as being immune-suppressive (Th2 or Treg) in origin [17].

Interestingly, we found a significant correlation between CD56 expression in TCs and the response to nivolumab immunotherapy. CD56, also known as *neural cell adhesion molecule* (NCAM), is a member of the immunoglobulin superfamily, and its expression is typically associated with NK cells [18,19]. However, it has been detected in other ICs, and it is aberrantly expressed in TCs of hematological malignancies (e.g., multiple myeloma and leukemia) as well as solid tumors (e.g., lung, ovarian, and renal cell cancers) [19]. In this context, a new role of CD56 in cancer and immune cell functioning has been observed. CD56 homodimerization between immune cells is implicated in communication and organization within the TME [20]. In addition, CD56 expression plays a role in cell cytotoxicity, providing the interaction between effector ICs and cancer cells via CD56-CD56 homophilic interactions (the so-called “*kiss of death*”) [20].

Molecular pathways in the TME have also been studied as prognostic factors, and the mTOR pathway is one of the most investigated ones in RCC patients [21,22]. The mTOR pathway regulates metabolism and thus the functions of numerous intra-tumor inflammatory cells and cancer cells [23]. We found that tumors in *responder* patients had higher expression of ph-mTOR. In fact, in addition to the regulation of the survival, differentiation, and migration of cancer cells, the activation of the mTOR pathway is associated with the immunoregulation of ICs and increased PD-L1 expression [23,24], which has been correlated with a higher benefit from immunotherapy in RCC patients [25]. Moreover, PD-L1 expressed in TCs may activate antiapoptotic signals, enhancing the PI3K–Akt-mTOR pathway and tumor-intrinsic glycolysis [26]. It can be assumed that the higher expression of ph-mTOR in the tumors of *responder* patients may be an indirect sign of the crucial role of PD-1/PD-L1 interaction; as a consequence, the action of nivolumab in these tumors may inhibit an essential manner of growth and proliferation. This hypothesis is in accordance with the numerically higher expression of PD-L1 in the tumors of *responder* patients.

Finally, we observed that the tumors of *non-responder patients* presented a higher grading (G3-G4). This result lies in the fact that the tumor grade is a well-known factor correlated with poor prognosis in patients with RCC [27]. Unexpectedly, we observed a higher probability of clear cell histology in the tumors of *non-responder* patients; we trust that the already planned molecular analyses will elucidate this finding.

In summary, responder patients seem to have a more immunological phenotype compared with non-responder patients, which seems to be correlated with the poor prognostic features generally associated with an angiogenetic phenotype. In this context, the different molecular profiles of mRCC are under investigation as potential drivers for the therapeutic choice. The BIONIKK trial is the first biomarker-driven trial that randomized mRCC patients to different treatments (immunotherapy monotherapy, immuno-combination, and TKI) according to tumor molecular characteristics according to a 35-gene expression mRNA signature [28]. What emerged from this study is that “immune-high” tumors benefit from immunotherapy monotherapy, “immune-low” tumors from immuno-combination, and “angio-high” tumors from TKI [28].

We acknowledge that among the limitations of the present analysis are its retrospective nature, the small size of the population, the lack of some tumor details (i.e., sarcomatoid differentiation), and the assessment of two different tumor specimens (primary tumors and metastases) with an underestimation of possible heterogeneity. However, the distribution of the primary tumor and metastasis specimens is similar between *responders* vs. *non-responders* (*p*-value *=* 0.93).

Moreover, another limitation regards the use of a less-used PD-L1 assay (Ventana PD-L1-SP263) compared with the PD-L1 IHC 22C3 pharmDx assay and the Dako PD-L1 staining, even though no standard PD-L1 assessment has been established in international guidelines and clinical practice.

The strength of our study was the attempt to identify molecular factors related to both the I-TME and tumors, the potential prediction of nivolumab treatment benefit, and the employment of easy-to-use and easily reproducible methods such as IHC.

Further analyses with a larger population are planned to confirm our IHC results. Moreover, the assessment of the gene expression profiles of angiogenic and immunosuppressive features in the two groups (*responders* and *non-responders*) is currently ongoing to assess the prognostic role of transcriptomic biomarkers and their correlation with the I-TME.

## 5. Conclusions

Many efforts have been made to investigate the prognostic and predictive role of the I-TME in cancer patients treated with immunotherapy, but no biomarkers have been established in clinical practice. This study aimed to identify potential differential I-TME and tumor features between *responder* and *non-responder* patients, which could help predict the response to immunotherapy. Further IHC and genomic analyses are planned to deeply examine the prognostic role of the I-TME.

## Figures and Tables

**Figure 1 cancers-15-02394-f001:**
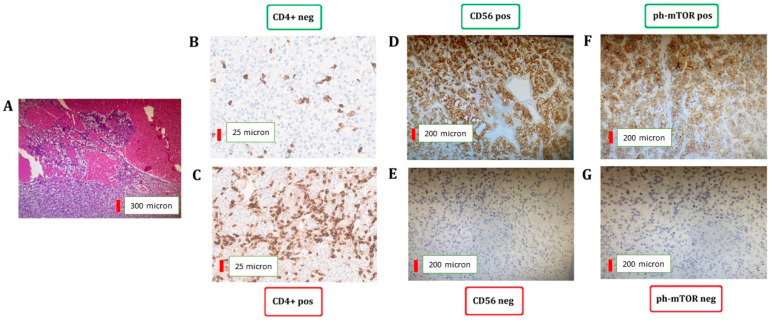
IHC assessment of CD4+ T lymphocytes and CD56 and ph-mTOR in TCs. Representative hematoxylin and eosin staining of clear cell renal cell carcinoma (**A**). CD4 (**B**,**C**), CD56 (**D**,**E**), and ph-mTOR (**F**,**G**) expression between *responders* (**B**,**D**,**F**) and *non-responders* (**C**,**E**,**G**).

**Table 1 cancers-15-02394-t001:** Patients’ and tumor samples’ characteristics.

Characteristic	N (%)			*p*-Value
Patients N = 84	All Patients	*Responders* (N = 37, 42.5%)	*Non-Responders* (N = 47, 57.5%)	
**Gender**				
Male	62 (73.8)	25 (67.6)	37 (78.7)	0.25
Female	22 (26.2)	12 (32.4)	10 (21.3)	
**Age** (median, range)	69, 27–84	73, 50–84	66, 27–82	** *0.004* **
**Histology**				
ccRCC	65 (77.4)	25 (67.6)	40 (85.1)	0.056
Other	19 (22.6)	12 (32.4)	7 (14.9)	
**Nephrectomy**				
Yes	72 (86.8)	34 (94.4)	38 (80.9)	0.070
No	11 (13.2)	2 (5.6)	9 (19.2)	
**IMDC score**				
Favorable	17 (20.2)	12 (32.4)	5 (10.6)	** *0.030* **
Intermediate	52 (61.9)	21 (56.8)	31 (66.0)	
Poor	15 (17.9)	4 (10.8)	11 (23.4)	
**Meet-URO score**				
1	18 (22.5)	8 (44.4)	10 (55.6)	0.21
2	28 (35.0)	14 (50)	14 (50)	
3	17 (21.3)	7 (41.2)	10 (58.8)	
4	10 (12.5)	4 (40.0)	6 (60.0)	
5	7 (8.7)	0 (0)	7 (100.0)	
**Treatment line**				
2nd line	60 (71.4)	26 (70.3)	34 (72.3)	0.84
≥3rd line	24 (28.6)	11 (29.7)	13 (27.7)	
**Samples N = 116**	**All samples**	**Samples of *Responders*** **(N = 55, 47%)**	**Samples of *Non-responders*** **(N = 61, 53%)**	
**Type of tumor sample**				
Primitive tumor	68 (59)	32 (58.2)	36 (59.0)	0.93
Metastasis	48 (41)	23 (41.8)	25 (41.0)	

N—number, ccRCC—clear cell renal cell carcinoma, and IMDC—International Metastatic RCC Database Consortium.

**Table 2 cancers-15-02394-t002:** IHC results of primary tumor and metastases samples between *responder* and *non-responder* mRCC patients.

Parameter (Cut-Off)	*Responders* N (%)	*Non-Responders* N (%)	OR (95% CI)	*p*-Value
**Histology**				
ccRCC	34 (61.8)	52 (85.3)	3.57 (1.46–8.71)	** *0.005* **
Other	21 (38.2)	9 (14.7)	1.00 (ref.)	
**Grading**				
1–2	17 (48.6)	10 (19.6)	1.00 (ref.)	
3	12 (34.3)	26 (51.0)	3.68 (1.30–10.40)	** *0.014* **
4	6 (17.1)	15 (29.4)	4.25 (1.25–14.50)	** *0.021* **
**CD3+ IC**				
Median (IQR)	90 (34–200)	45 (25–210)	-	0.77
<40	15 (27.3)	27 (44.3)	1.00 (ref.)	0.059
≥40	40 (72.7)	34 (55.7)	0.47 (0.22–1.03)	
**CD8+ IC**				
Median (IQR)	100 (25–150)	105 (25–139)	-	0.76
**CD4+ IC**				
Median (IQR)	45 (12–70)	60 (15–88)	-	0.22
<70	41 (74.6)	32 (52.5)	1.00 (ref.)	** *0.015* **
≥70	14 (25.5)	29 (47.5)	2.65 (1.21–5.83)	
**CD8+/CD4+ ratio** Median, IQR	1.74 (0.54–3.71)	1.20 (0.32–2.39)	-	0.084
**Peri/intra-tumoral T cells**				
Absent	24 (43.6)	26 (42.6)	1.00 (ref.)	0.91
Present	31 (56.4)	35 (57.4)	1.04 (0.50–2.18)	
**CD56 TC**				
Median (IQR)	0 (0–40)	0 (0–10)	-	0.23
<40	40 (72.7)	54 (88.5)	1.00 (ref.)	** *0.035* **
≥40	15 (27.3)	7 (11.5)	0.35 (0.13–0.93)	
**CD15 TC**				
Median (IQR)	1 (0–10)	1 (0–5)	-	0.70
<30	48 (87.3)	48 (78.7)	1.00 (ref.)	0.23
≥30	7 (12.7)	13 (21.3)	1.86 (0.68–5.06)	
**CD68 TC**				
Median (IQR)	0 (0–40)	0 (0–10)	-	0.77
**ph-mTOR TC**				
Median (IQR)	20 (10–70)	15 (0–70)	-	0.25
<15	16 (29.1)	30 (49.2)	1.00 (ref.)	** *0.029* **
≥15	39 (70.9)	31 (50.8)	0.42 (0.20–0.91)	
**PD-L1 TC/IC**				
Median (IQR)	3 (0–10)	0 (0–5)	-	0.46
<10	40 (72.7)	51 (83.6)	1.00 (ref.)	0.16
≥10	15 (27.3)	10 (16.4)	0.52 (0.21–1.29)	

N—number of patients, OR—odds ratio, CI—confidence interval, ccRCC—clear cell renal cell carcinoma, ref.—reference, IQR—interquartile range, TC—tumor cell, and IC—immune cell.

## Data Availability

All the data are accessible and available upon request from the corresponding author.

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
