# Peer review of "Characterization of Tumor and Immune Tumor Microenvironment of Primary Tumors and Metastatic Sites in Advanced Renal Cell Carcinoma Patients Based on Response to Nivolumab Immunotherapy: Preliminary Results from the Meet-URO 18 Study"

_cancers, 2023, doi:10.3390/cancers15082394_

Round 1

Reviewer 1 Report

Summary: a retrospective study to find the immunologic predictive factors of response to nivolumab in patients with metastatic RCC. 

Comments: 

1. The main title reflects the evaluation of I-TME in metastatic RCC. This title can reflect that the study has been done in metastatic tissue. However, 59% of the tissue samples in this study were primary tumors. It is suggested to revise the main title to impede this misunderstanding. For example, the following choice is suggested:

"Characterization of tumor and immune tumor microenvironment of primary and metastatic renal cell carcinoma based on response to nivolumab immunotherapy: preliminary results from the Meet-URO 18 study. "

2. Introduction: "Prognostic and/or predictive factors, under investigation in mRCC patients treated with immunotherapies, include peripheral blood inflammatory indices, clinical factors, and tumor microenvironment (TME) biomarkers [3,4]". Recent evidence has put forward the role of mitochondrial metabolism in response to anti-PD-1 immunotherapies (https://pubmed.ncbi.nlm.nih.gov/36469835/). Houshyari et al. demonstrated that increased immune cells' mitochondrial biogenesis can enhance the response to anti-PD1 immunotherapies. It is recommended the authors mention this (and similar) concepts to improve the background of the study. 

3. Study population: it has been noted that cytologically diagnosis of RCC is an inclusion criteria. However, in this case, doing IHC is not practical. Please outline this issue. 

4. Table 2. please explain the reason to evaluate the factor PD-L1 TC/IC. Based on the practice, we usually evaluate CPS or TPS to determine the clinical response to anti-PD1 immunotherapy. However, we see that these factors have not been evaluated. The authors should evaluate the effect of CPS (or TPS) on response to nivolumab. 

5. The study found that high-grade tumors responded weaker to nivolumab. This finding can contrast with the literature, where high-grade tumors (with more genetic instability) respond better to the anti-PD1 antibodies. It is suggested the authors discuss this controversy in the Discussion section.  

Author Response

We would like to thank the reviewer n. 1 for the important and useful comments. Here, there are the answers point by point:

  1. Since the clinical assessment of RCC patients has been made in their metastatic setting, we have modified the title: "Characterization of tumor and immune tumor microenvironment of primary tumors and metastatic sites of advanced renal cell carcinoma patients based on response to nivolumab immunotherapy: preliminary results from the Meet-URO 18 study"
  2. We have included a sentence with the reference in the background section
  3. We have modified the sentence
  4. The PD-L1 is a well-known negative prognostic factor but a clear predictive value has not been established for mRCC patients. It is not considered in international guidelines nor used in clinical practice to assess the clinical response. Moreover, many different types of PD-L1 antibodies have been used in the different trials (Dako CKM 025/CKM 214, 22C3 KN426/CLEAR, SP263 JAVELIN Renal 101 as our analysis) and none is considered a standard tool. In consideration of the abovementioned reasons, we have chosen to use PD-L1 on IC and TC to assess both the tumor and immune compartments of the tumor microenvironment. We have added the limits of our study regarding the PD-L1 assessment in the discussion section.
  5. We know that the sarcomatoid histology (most but not all G4) is associated with a great benefit with immuno-based therapy compared with target therapy. However, we have no direct comparison of the different efficacy of immunotherapy in sarcomatoid vs non-sarcomatoid patients. Moreover, to our knowledge, data on the differential response according to G3 are not available.

Moreover, the English has been checked and some modification have been made.

Reviewer 2 Report

Dear authors:

The assessment of TME after nivolumab treatment in clinic has been described properly with middle-size data. While the experiments perfectly reflect the fact that nivolumab helps T-cell infiltration and changed the TME significantly, some other data did not show promising outcomes. 

I believe that as you mentioned in the last, the data size should be expanded, and more transcriptome data needs to be elucidated. This is a great conclusion to acknowledge the limitation.

I suggest that if possible, add the following assessments to the TME:

a. cytokine study on the tumor sample

b. check Tumor cell APC and MHC level

c. vascular growth of the tumor

For a and b are valuable to understand the nivolumab effect on tumor elimination, and c is a possible reason why non-responders failed the treatment. Research found the VGFR in tumor may help drug delivery like nivolumab while less vascular growth in tumor may lead to failure of drug delivery. 

Author Response

We would like to thank the reviewer n. 2 for the important suggestions on further and deeper analyses. The expanded data for IHC analyses and the transcriptome analyses are currently ongoing and we will consider the suggested assessments of the TME to better understand the nivolumab effect and delivery. Moreover, the English has been checked and some modification have been made.

Reviewer 3 Report

Overall, well designed study. I have a few minor comments only. 

1. Can we use the Meet URO score along with IMDC criteria? 

2. What is the clinical significance of differential expression amongst the responders versus nonresponders? Are there any studies using a similar type of information to design trials, if not, any proposed hypothesis to use this information?

3. What is the median of cycles received?

Author Response

We would like to thank the reviewer n. 3 for the important comments. Here, there are the answers point by point:

  1. We have updated the Table 1 with the Meet-URO score data
  2. We have added a paragraph on the significance of the differential expression between responders and non-responders and on the BIONIKK trial.
  3. Unfortunately, we have no data on the number of the cycles received. The clinical division of the mRCC groups assessed is based on PFS and not on the amount of therapy received.

Moreover, the English has been checked and some modification have been made.

Reviewer 4 Report

There is a big different result to compare of Responders v.s. Non-responders ccRCC% from patient samples (p=0.056 n=84) and tumor samples (0.005 N=116). It should be the same or similar %, otherwise one of them (patient samples or tumor samples) will has the bias to be representing for the real drug responses of all the RCC. Maybe this is why PDL1 have no significant different between Responders v.s. Non-responders (p=0.16). It better to put the one patient with only same one tumor sample to assay again of the Table 2. IHC results.

In addition, it’s needed to put the representative IHC staining results of CD4+, CD56+, total-mTOR and phospho-mTOR of Responders v.s. Non-responders.

Author Response

We would like to thank the reviewer n. 4 for the important comments. The Table 1 shows the distributions of patients’ characteristics between responders vs non-responders so the aim is descriptive, while the Table 2 shows the odds ratios of the IHC parameters. According to the distributions of patients’ characteristics, the distribution of the type of specimen (primitive tumor and metastatic sites) is similar between responders and non-responders which is a strength of the analysis. About that, a sentence is added in the discussion section.

Regarding the histology, the difference between the two groups in Table 1 is borderline (p value = ) but the percentage distribution is very similar to that reported in Table 2 where the p value is significant most likely due to the increase of the sample size. For this preliminary analysis, we prefer to use all samples available because they are equally distributed between the two clinical groups and to maintain a higher sample size and reduce the probability of running into the beta error. For further analysis on a higher number of patients enrolled, we will make the analysis "one specimen – one patient" as advised by the reviewer.

The figure on the representative IHC staining results of CD4+, CD56+ and phospho-mTOR has been added. Unfortunately, we have not analysed total-mTOR.

Round 2

Reviewer 1 Report

All my comments are addressed. I have no more comments. 

Author Response

We are very grateful to the reviewer for the helpful comments and suggestions.

Reviewer 4 Report

1. “patients (pts)” were only used in abstract but were no appearing in the main article. It should not be necessary to use abbreviation in the abstract.

2. In Figure 1, the IHC of each figure should be added the scale bar and magnification.

3. In Line 187-188, “CD4 (A,B), CD56 (D,E) and ph-mTOR (F,G) expression between responders (B,D,F) and non-responders (C,E,G).” should be “CD4 (B,C), CD56 (D,E) and ph-mTOR (F,G) expression between responders (B,D,F) and non-responders (C,E,G).”

Author Response

We really thank reviewer n. 3 for the useful suggestions. We have modified the text and the figure as requested.